# Perioperative Chemo/Immunotherapies in Lung Cancer: A Critical Review on the Value of Perioperative Sequences

**DOI:** 10.3390/curroncol32070397

**Published:** 2025-07-10

**Authors:** Thoma’ Dario Clementi, Francesca Colonese, Stefania Canova, Maria Ida Abbate, Luca Sala, Francesco Petrella, Gabriele Giuseppe Pagliari, Diego Luigi Cortinovis

**Affiliations:** 1Medical Oncology Unit, Fondazione IRCCS San Gerardo dei Tintori, 20900 Monza, Italy; francesca.colonese@irccs-sangerardo.it (F.C.); stefania.canova@irccs-sangerardo.it (S.C.); mariaida.abbate@irccs-sangerardo.it (M.I.A.); luca.sala@irccs-sangerardo.it (L.S.); g.pagliari1@campus.unimib.it (G.G.P.); diegoluigi.cortinovis@irccs-sangerardo.it (D.L.C.); 2Medicine and Surgery Department, Milano Bicocca University, 20126 Milan, Italy; 3Department of Thoracic Surgery, Fondazione IRCCS San Gerardo dei Tintori, 20900 Monza, Italy; francesco.petrella@irccs-sangerardo.it

**Keywords:** perioperative in lung cancer, resectable/early-stage NSCLC, chemo-immunotherapies for NSCLC, neoadjuvant ICIs

## Abstract

The treatment paradigm for rNSCLC has undergone a significant evolution with the integration of immunotherapy into perioperative strategies. Landmark trials such as IMpower010 and PEARLS have demonstrated DFS benefits with adjuvant atezolizumab and pembrolizumab, particularly in tumors ≥ 4–5 cm and/or with PD-L1 expression ≥ 1%. Concurrently, neoadjuvant ICI-based regimens, as evidenced by CheckMate 816, NADIM II, and AEGEAN, have shown substantial MPR rates and improvements in EFS, prompting recent FDA approvals. Emerging data from perioperative trials, such as KEYNOTE-671, RATIONALE-315, Neotorch, and CheckMate 77T, underscore the synergistic efficacy of chemo-immunotherapy in enhancing long-term outcomes. The integration of ICIs both pre- and postoperatively is redefining rNSCLC management, with perioperative immunotherapy poised to become a cornerstone in early-stage disease.

## 1. Introduction

Despite great changes in lung cancer treatments occurring over the past decade, non-small-cell lung cancer (NSCLC) stands as the primary cause of cancer-related mortality globally, with approximately 1.8 million lung cancer deaths recorded in 2024 [1]. Regrettably, most patients (60–70%) are diagnosed at advanced stages, precluding curative treatment [2]. Yet, even among those with early-stage disease, NSCLC presents a significant clinical challenge due to its proclivity for distant metastasis and early recurrence. Consequently, enhancing outcomes for patients with resectable NSCLC (rNSCLC) has become a paramount focus of multidisciplinary research in recent years.

The landscape of early-stage NSCLC is evolving, emerging into two distinct population groups with distinct clinical characteristics and therapeutic options. According to the eighth edition of the tumor–node–metastasis staging classification (TNM) [3], in very early-stage disease (stage IA1-IA2), the field of scientific debate revolves around optimizing resection extent while maintaining oncologic integrity. Conversely, in patients with stage IB-IIIA disease, the risk of recurrence is considerably higher, requiring a multimodal treatment approach that can include systemic therapy, radiation, and surgery for maximal efficacy.

The current best-in-class chemotherapy for early-stage NSCLC is adjuvant platinum despite the high recurrence and mortality rates as it does increase the 5-year overall survival (OS) rate by around 5% when compared with surgery alone [4,5].

However, in recent years, the treatment paradigms for advanced and metastatic NSCLC has been significantly altered for the better using of immune checkpoint inhibitors (ICIs) [6,7,8].

As a result, immunotherapy has also been explored in resectable NSCLC with favorable results in many clinical trials. However, many gray areas are still a matter of debate.

In this review, we provide an overview of the main relevant clinical trials in the setting of neoadjuvant–preoperative–adjuvant treatment using iCIs in early-stage NSCLC, excluding patients with mutations of EGFR/ALK. We also focus on the gray areas that are controversial and need to be better investigated and clarified.

## 2. Research Methods

A comprehensive literature search was conducted using the PubMed, Google Scholar, EMBASE, Cochrane Library, and clinicaltrials.gov databases. These sources were selected for their capacity to deliver robust scientific evidence, including phase II and phase III trials, systematic reviews, and pertinent clinical and preclinical research. We included articles published in English from 1 January 2003 to 1 January 2025. Additionally, our search strategy employed key terms such as “Perioperative in Lung Cancer,” “Resectable NSCLC,” “Early-stage NSCLC,” “Chemo-immunotherapies NSCLC,” and “Neoadjuvant ICIs” to ensure comprehensive coverage of the subject matter.

## 3. Adjuvant Setting

The traditional approach for rNSCLC has long involved surgery followed by adjuvant platinum-doublet chemotherapy (PDC) administration for patients deemed high-risk for recurrence. Typically, NCCN guidelines advise against adjuvant PDC for stage IA disease according to the eight edition of the American Joint Committee on Cancer (AJCC), citing data from the Lung Adjuvant Cisplatin Evaluation (LACE) meta-analysis that survival rates are correlated with a potential decline [5]. However, despite the absence of complete consensus, for those with stage IB disease, NCCN guidelines lean towards considering adjuvant PDC for tumors showing high-risk features [9].

In 2008 the IALT trial [10] and the LACE meta-analysis shows that the use of adjuvant platinum-based chemotherapy in rNSCLC patients increases the 5-year overall survival (OS) rate by around 5.4% when compared with surgery alone.

Using immunotherapy in the adjuvant setting is based on the idea that it could benefit patients who have an inadequate response to neoadjuvant chemo-immunotherapy, especially in the presence of minimal residual disease (MRD). Emerging evidence indicates that tumor excision might unfavorably alter the natural history of MRD, thereby increasing the risk that micrometastases will proliferate and evade the immune system [11,12].

The combination of immunotherapy plus chemotherapy synergically acts to enhance the control of the disease. Since chemotherapy disrupts tumor cells and generates neoantigen exposure, this could improve the outcomes [13].

Currently, two adjuvant immunotherapy regimens are approved for treating rNSCLC. Atezolizumab is approved following PDC for resected stages II to III NSCLC with a PD-L1 TPS expression of ≥1% (>50% in some countries), while Pembrolizumab has also been approved for stage IB to IIIA rNSCLC, irrespective of PD-L1 TPS expression.

The IMpower010 study [14] investigated adjuvant ICIs within completely resected stage IB–IIIA NSCLC and found that adjuvant platinum-based chemotherapy was the most effective. Following this, a randomized cohort of 1005 patients received adjuvant atezolizumab for a total of 16 cycles or best supportive care (BSC). The primary endpoint, disease-free survival (DFS), was significantly improved in the experimental arm (adjuvant atezolizumab) in the subgroup of the stage II–IIIA population with PDL1 expression ≥ 1% (median NE vs. 35.3 months; HR: 0.66, 95% CI: 0.50–0.88; *p* = 0.0039) and in all stage II-IIIA patients (median 42.3 vs. 35.3 months; HR: 0.79, 95% CI: 0.64–0.96; *p* = 0.020). However, in the intention-to-treat (ITT) population (stage IB–IIIA), the hazard ratio (HR) for DFS was 0.81, not surpassing the statistical significance threshold. The DFS benefit of atezolizumab was specifically seen in patients with PD-L1 expression ≥ 50% (HR: 0.43 and 95% CI: 0.27–0.68) compared to those with PD-L1 expression 1–49% (HR: 0.87 and 95% CI: 0.60–1.26), suggesting that the disease-free survival benefit of adjuvant atezolizumab may be limited to patients with PD-L1 ≥ 50%.

The likelihood of recurrence is closely tied to the cancer stage, making the risk-to-benefit assessment of systemic therapy more advantageous in earlier stages. The IMPower010 trial, which evaluated adjuvant atezolizumab following cisplatin-based chemotherapy, demonstrated a significant DFS benefit in patients with tumors measuring at least 5 cm or with lymph node involvement, but not in those with tumors between 4 and 5 cm. This benefit was particularly evident in tumors expressing PD-L1. On the other side, the PEARLS trial found that adjuvant pembrolizumab conferred a survival advantage in tumors of at least 4 cm, irrespective of nodal involvement or PD-L1 expression.

Subsequently, thanks to the IPOWER010 trial, the FDA approved pre-operative atezolizumab in resectable stage II to IIIA NSCLC with a PD-L1 expression of 1% or greater [15,16].

On the other side, the randomized, triple-blind, phase 3 PEARLS trial [17] found at the second interim analysis that adjuvant pembrolizumab, compared to the placebo, conferred a survival advantage in tumors of at least 4 cm, irrespective of nodal involvement or PD-L1 expression (n = 1177; HR, 0.76 [95% CI, 0.63–0.91]; *p* = 0.0014).

Unlike in locally advanced, unresectable NSCLC or in the perioperative setting, adjuvant durvalumab, after R0 resection, did not lead to a significant improvement in DFS outcomes in rNSCLC. In fact, the LBA48 CCTG BR.31 [18] randomized 1415 patients with stage IB, II, or IIIA R0 NSCLC to receive durvalumab vs. the placebo. The primary endpoint, DFS in patients with PD-L1 TPS ≥ 25% excluding common EGFR or ALK mutations, was not reached.

The lack of DFS improvement contrasts with findings from trials such as the IMpower010 and PEARLS trials, highlighting the complex and context-specific nature of ICI activity in adjuvant therapy for early-stage NSCLC.

## 4. Neoadjuvant Setting

The introduction of neoadjuvant platinum-based chemotherapy for rNSCLC has historically been underutilized. However, randomized trials have demonstrated its efficacy, leading to its inclusion in the NCCN guidelines as a viable option for patients with early-stage NSCLC. Studies have shown that neoadjuvant chemotherapy can improve OS rates by approximately 5% at five years, a benefit comparable to adjuvant chemotherapy.

With the advent of immunotherapy, ICIs have been studied both as a monotherapy and in combination with chemotherapy and radiotherapy in the neoadjuvant setting (Table 1 and Table 2), operating on the hypothesis that the presence of an unaltered tumor immune microenvironment may lead to higher responses using immune checkpoint inhibitors. [19].

Among the strategies employed, some trials combined PD-1/PD-L1 inhibitors with CTLA-4 antagonists (as observed in NEOSTAR, especially in the NEOSTAR platform phase II trial), integrated them with stereotactic body radiation therapy (as conducted by Altorki et al. 2021) [21], or introduced novel agents into the treatment regimen (as exemplified by NEOCOAST). Notably, adverse event profiles remained consistent with established ICI safety parameters, with incidences of grade 3–5 adverse events ranging from 10% to 30% [4,5,6,7,8,9,10,11,12,13,14,15,16,17,18,19,20,21,22,23,24,25,26,27,28,29,30,31,32].

Research on neoadjuvant monotherapy with anti-PD-(L)1 agents has demonstrated major pathological response (MPR) rates ranging from 6.7% to 45%.

In the multicenter Lung Cancer Consortium 3 (LCMC3) study, the neoadjuvant application of atezolizumab yielded a 19% MPR rate, aligning with the efficacy observed in prior studies utilizing cisplatin-based neoadjuvant chemotherapy.

In 2020, a neoadjuvant investigation of the PD-1 inhibitor sintilimab in a cohort of 37 Chinese patients with rNSCLC reported a notable 41% MPR rate [33]. While a Chinese phase 2 study, TD-FOREKNOW, which enrolled 88 patients, demonstrated that three cycles of camrelizumab plus neoadjuvant chemotherapy versus neoadjuvant chemotherapy alone can significantly increase this rate to 65.1% [34].

The phase II randomized NEOSTAR trial examined the neoadjuvant nivolumab alone and in combination with ipilimumab in patients with resectable NSCLC. The trial documented a 38% MPR rate when using the combination therapy in 21 patients, with a substantial proportion of these responses being a pathological complete response (pCR). Further, the randomized phase 2 NEOSTAR trial evolved into a platform trial of sequential, single-center, single-arm, phase 2 studies following the high activity of the neoadjuvant PDC when combined with ICIs. The results reflected the initial hypotheses, showing an MPR rate of 50% (11/22) and 32.1% (7/22), respectively.

The combination of chemotherapy’s cytotoxic effects with the immune-modulating capabilities of ICIs aims to amplify therapeutic efficacy reciprocally. Therefore, it potentially improves survival rates and enhances the quality of life for patients.

In 2022, the FDA sanctioned the use of the neoadjuvant nivolumab alongside platinum-doublet chemotherapy for rNSCLC, following the positive outcomes of the phase III CheckMate 816 trial [29]. This led the National Comprehensive Cancer Network to endorse this regimen for patients with stage IB to IIIA or IIIB (specifically T3 and N2) NSCLC.

CheckMate 816 was a phase 3, open-label trial that randomized adults with resectable stage IB (≥4 cm) to IIIA NSCLC (AJCC v7), an ECOG performance status of ≤ 1, and no known EGFR or ALK alterations to receive either the neoadjuvant nivolumab plus platinum-based chemotherapy or chemotherapy alone, both administered every three weeks for three cycles prior to surgery. The study provided robust evidence that the addition of nivolumab significantly improved EFS (31.6 months vs. 20.8 months with platinum-based chemotherapy alone) and MPR (24.0% vs. 2.2% of patients, respectively). Notably, CheckMate 816 remains the only neoadjuvant-only phase 3 immunotherapy trial to show a statistically and clinically meaningful OS advantage at 5 years, with OS rates of 65% in the combination arm compared to 55% in the platinum-based chemotherapy arm. Moreover, patients achieving a pCR experienced a striking OS benefit at 5 years (95% vs. 56%) [35].

## 5. Perioperative Setting

Perioperative immunotherapy, spanning both neoadjuvant and adjuvant treatment phases, is gaining momentum in managing rNSCLC, presenting both challenges and potential benefits.

While amalgamating neoadjuvant and adjuvant immunotherapy complicates deciphering individual treatment contributions, it may offer advantages, particularly for patients with inadequate responses to neoadjuvant therapy. Diverse outcomes from various perioperative trials, such as NADIM II, AEGEAN, Neotorch, KEYNOTE-671, Checkmate 77T, and RATIONALE-15, underscore the evolving landscape of NSCLC treatment [36,37,38,39,40,41] (Table 3).

The NADIM II trial evaluated neoadjuvant nivolumab alongside PDC followed by adjuvant nivolumab, juxtaposed with neoadjuvant PDC monotherapy, in stage IIIA or stage IIIB rNSCLC. The results were nothing short of remarkable, unveiling a substantially elevated pCR rate of 37% with perioperative nivolumab compared to a 7% rate with neoadjuvant PDC alone (RR 5.34, *p* = 0.002).

Simultaneously, the AEGEAN trial is a phase III, placebo-controlled study evaluating the effects of neoadjuvant durvalumab combined with PDC, followed by adjuvant durvalumab, compared to a regimen of neoadjuvant PDC plus placebo and adjuvant placebo, in patients with stage IIA to IIIB rNSCLC. After excluding tumors with EGFR and ALK mutations and with a median follow-up of 11.7 months, the median EFS had yet to be reached in the durvalumab arm, compared to 24.9 months in the PDC plus placebo arm. The inclusion of durvalumab resulted in a 32% reduction in the risk of disease progression, recurrence, or death (HR: 0.68). Additionally, the pCR rate following neoadjuvant therapy was significantly higher in the durvalumab group (17.2%) compared to the control group (4.3%), demonstrating a notable 13% difference (*p* < 0.001). While these findings echo the benefits observed with neoadjuvant chemoimmunotherapy in the Checkmate 816 trial, the added advantage of the adjuvant durvalumab therapy remains ambiguous, as the overall outcomes were largely comparable.

CheckMate 77T was a phase 3, randomized, double-blind trial, enrolling patients with stage IIA to IIIB rNSCLC to receive neoadjuvant nivolumab plus chemotherapy or neoadjuvant chemotherapy plus the placebo every 3 weeks for four cycles. This was followed by surgery and adjuvant nivolumab or the placebo every 4 weeks for 1 year, which resulted in significantly longer EFS than chemotherapy. Specifically, the percentage of patients with 18-month EFS was 70.2% in the nivolumab group and 50.0% in the chemotherapy group (HR for disease progression or recurrence, abandoned surgery, or death, 0.58; 97.36% confidence interval [CI], 0.42 to 0.81; *p* < 0.001). A pCR occurred in 25.3% of the patients in the nivolumab group and in 4.7% of those in the chemotherapy group (odds ratio, 6.64; 95% CI, 3.40 to 12.97).

The phase III RATIONALE-315 study investigated tislelizumab combined with chemotherapy in Chinese patients with Stage II-IIIA rNSCLC. Patients received 3–4 cycles of preoperative tislelizumab at the dose of 200 mg or a placebo (both every 3 weeks) combined with PDC, followed by up to 8 cycles of adjuvant tislelizumab with a double dose of 400 mg or the placebo (both every 6 weeks) postoperatively. The arm with tislelizumab plus chemotherapy achieved a considerable improvement in pCR rates (40.7% vs. 5.7%) and MPR rates (56.2% vs. 15.0) compared with chemotherapy alone. The benefit of the combination was demonstrated in all patients regardless of the histology or the PD-L1 expression level. Definitive surgery was offered in 84.1% in the tislelizumab arm and 76.2% in the placebo arm. The safety profile was consistent with the other immunotherapeutic plus chemotherapy.

Moreover, the Neotorch trial, presented at the 2023 ASCO Annual Meeting, showcased the efficacy of the anti-PD-1 agent toripalimab. When administered in combination with PDC for three preoperative cycles, followed by 1 postoperative cycle of PDC and up to 13 cycles of toripalimab monotherapy, this regimen was associated with a 60% reduction in the risk of EFS events (HR 0.40, *p* < 0.001) in patients with operable stage II to III NSCLC, according to the eight edition of the AJCC.

The KEYNOTE-671 trial further emphasized the potential of immunotherapy in this setting. Patients with resectable stage II to IIIB NSCLC were randomized to receive either neoadjuvant pembrolizumab or the placebo alongside PDC for 4 cycles, followed by adjuvant pembrolizumab or the placebo for up to 13 cycles. The results at the second interim analysis at 36 months [48] demonstrated significant improvements in the primary endpoints of OS of 71% (95% CI: 66–76) in the pembrolizumab group and 64% in the placebo group (hazard ratio: 0.72 [95% CI 0.56–0.93]), and the median EFSl was 47.2 months (95% CI: 32·9 to not reached) in the pembrolizumab group and 18·3 months (14.8–22.1) in the placebo group (HR: 0·59 [95% CI 0·48–0·72]). These outcomes have led to the recent FDA approval of perioperative pembrolizumab for rNSCLC.

## 6. Discussion

This review has examined recent advances in the treatment of rNSCLC. The recent FDA approval of three new regimens (nivolumab, pembrolizumab, and atezolizumab) for rNSCLC has firmly established immune checkpoint inhibitors as a new standard of care for patients with resectable stage IB-IIIA NSCLC. Nonetheless, important challenges remain. In the following discussion, we try to explore these open questions to outline future directions for research and clinical practice.

### 6.1. Aiming at Customizing the Treatment

At present, there are no direct comparisons between neoadjuvant vs. adjuvant chemoimmunotherapy, and the relative advantages and disadvantages of either approach are unclear. Administering systemic therapy before surgery offers the opportunity to observe the tumor response, potentially guiding subsequent therapeutic approaches [49].

This strategy holds promise due to the relatively homogeneous nature of tumor neoantigens before exposure to the selective pressures exerted by cancer treatments [50].

Additionally, there is the potential for heightened immune responses mediated by tumor-specific CD8+ T cells by preserving the integrity of the primary tumor and lymphatic system, as opposed to the methodology in postoperative or postradiotherapy contexts. Upon activation, these T cells can proliferate extensively, migrate peripherally, and infiltrate not only the primary tumor but also sites of micrometastatic disease. Subsequently, the persistent presence of tumor-specific CD8+ T cells following local therapy may contribute to mitigating the risk of recurrence after definitive treatment. Moreover, the blockade of PD-1/PD-L1 pathways may further stimulate the reactivation and expansion of tumor-specific T cells. This effect is partly attributed to the presence of dendritic cells expressing PD-L1 within the tumor microenvironment, which play pivotal roles in immune checkpoint inhibition [51].

Preclinical studies also shed light on the unique advantages of administering immune checkpoint inhibitors in the neoadjuvant rather than adjuvant setting. In murine models of resectable NSCLC, neoadjuvant treatment with nivolumab alone or in combination with ipilimumab significantly prolonged survival and is correlated with increased infiltration of tumor-infiltrating lymphocytes (TILs) compared to adjuvant administration [52].

Adjuvant chemotherapy is effective for eliminating postoperative minimal residual disease and combating the immunosuppressive environment, while adjuvant immunotherapy synergizes with postoperative PDC, enhancing the antitumor immune response [53].

A recent meta-analysis including eight trials indicated a significant benefit from neoadjuvant ICI chemotherapy. The benefit consisted of improved 2-year EFS (HR, 0.57; 95%CI, 0.50–0.66; *p* < 0.001) and an increased pCR rate (RR, 5.58; 95%CI, 4.27–7.29; *p* < 0.001) in the experimental vs. control treatment arms [54].

Specifically, only one of these trials, the Keynote-671 trial [36], had OS as one of the primary endpoints that was met. The other studies included in the review [19,31,35,37,38,39] did not include adjuvant ICI therapy and investigated pCR or MPR and/or EFS as primary endpoints with different therapeutic strategies such as administering three or four neoadjuvant ICI-chemotherapy cycles or using either carboplatin- or cisplatin-based chemotherapy.

However, EFS and pCR are considered valuable indicators for OS [55]. The authors concluded that three cycles of neoadjuvant platinum-based ICI-chemotherapy were associated with a meaningful improvement in 2-year EFS and pCR. The benefit was reported in all patients. However, some subgroups had better HRs and 95% CIs for 2-year EFS, such as patients with stage III disease (when compared to those with stage IB-II). Moreover, patients who were negative for PD-L1 were at a higher risk of relapse (HR, 0.75; 95% CI, 0.62–0.91) when compared with patients with low (HR, 0.61; 95% CI, 0.37–0.71) or high PD-L1 (HR, 0.40; 95%CI, 0.27–0.58) (*p* = 0.005).

Finally, the final report from the NADIM study reinforced that attaining a pCR after neoadjuvant therapy serves as a robust predictor of improved long-term survival [56].

Since head-to-head randomized trials have not been performed to answer the question of whether giving PD-1 inhibition both before and after surgery provides superior outcomes versus solely before resection, this topic remains controversial and a matter of debate.

In 2024 only one analysis, using individual patient-level data from the CheckMate 816 and CheckMate 77T trials, compared neoadjuvant nivolumab + platinum-based chemotherapy versus perioperative nivolumab in rNSCLC [57]. The analysis showed that the addition of at least one dose of adjuvant nivolumab was associated with a 40% reduction in the risk of recurrence. This benefit was more pronounced in patients with PD-L1 < 1% and, notably, was also consistent in patients without a pCR [29].

However, there is no conclusive answer; however, a better selection of patients to customize the treatment is essential to define who is at a higher risk of relapse. Some promising biomarkers for possible escalation or de-escalation therapy include the quantitative or qualitative assessment of circulating DNA.

Many data suggest a possible role in using the circulating tumor DNA (ctDNA) as a biomarker in lung cancer. In rNSCLC, trials like NADIM, AEGEAN, and Checkmate 816 have shown that lower ctDNA levels before and after neoadjuvant therapy correlate with better survival outcomes. The LCMC3 and IMpower010 trials further highlight ctDNA’s role in predicting treatment responses and MRD. Ongoing studies continue to explore the integration of ctDNA as a biomarker in early-stage NSCLC, offering insights into the tumor immune microenvironment and its response to neoadjuvant therapies [58].

Moreover, the DYNAMIC study [59] demonstrated that ctDNA measurements taken just three days post-surgery could already predict a patient’s prognosis, showing that individuals with undetectable ctDNA had significantly longer recurrence-free survival compared to those with detectable levels. However, other experts suggest that the optimal window for initial ctDNA MRD testing may be 1–2 weeks post-surgery to ensure more reliable results [60].

In a related study by Jung et al. [61], researchers examined ctDNA levels at various points—pre-surgery, four weeks post-surgery, and at regular intervals over five years. Patients were stratified according to their circulating tumor DNA (ctDNA) status, showing that those with undetectable ctDNA after surgery had significantly higher three-year disease-free survival (DFS) rates compared to those with persistent ctDNA. These results highlight the potential role of ctDNA-based MRD testing for dynamic risk stratification and early recurrence detection. However, its implementation in clinical practice still requires careful consideration of cost-effectiveness and the need for standardized testing protocols.

Finally, host-related markers such as the gut microbiome are under investigation. The microbiome plays a key role in the modulation of inflammation and in the immune response. Therefore, it may predict the prognosis and effectiveness of ICIs in NSCLC treatment. In the NEOSTAR trial baseline fecal microbiota in patients with MPR was enriched with beneficial taxa, such as Akkermansia, and displayed reduced abundance of pro-inflammatory and pathogenic microbes.

The topic of escalation or de-escalation therapy is crucial to the accessibility of immunotherapy across the countries due to its cost and possible side effects, as well as long-term or permanent consequences.

### 6.2. Patient’s Compliance

Limits of the adjuvant approach include longer durations of systemic therapy that can be associated with lower adherence rates compared to shorter neoadjuvant regimens. For instance, while 94% of patients completed neoadjuvant chemoimmunotherapy in the Checkmate 816 trial, lower completion rates were observed in IMpower010 where only 65% of patients received the full 16 cycles of adjuvant immunotherapy.

In the paucity of evidence about the safety of immune checkpoint blockade in the neoadjuvant and adjuvant setting, a systematic review and meta-analysis of 28 randomized clinical trials in rNSCLC showed that neoadjuvant ICIs were not associated with a significant increase in treatment-related deaths or grade 3–4 immuno-related adverse events, suggesting a favorable safety profile (OR: 1.11, 95% CI: 0.38–3.29, and *p* = 0.84; OR: 1.17, 95%: CI 0.90–1.51, and *p* = 0.23, respectively). Conversely, adjuvant therapy was associated with a higher risk of both treatment-related mortality (OR: 4.02; 95% CI: 1.04–15.63; *p* = 0.044) and severe toxicity (OR: 5.31; 95% CI: 3.08–9.15; *p* < 0.0001) [62].

These insights align with current guidelines and practice paradigms, emphasizing the importance of balancing therapeutic benefits with potential risks.

### 6.3. Concept of Resectability and Operability

The considerable data being generated by trials in neoadjuvant and adjuvant immunotherapy for patients with resectable lung cancer has created a wealth of various possibilities and accompanying challenges regarding best practice. In particular, the criteria of resectability for stage III.

A recent consensus [63] aimed to clarify some gray areas for the selection of the candidates for neoadjuvant/preoperative treatment. Best practice might be well served by creating a multidisciplinary tumor board to define optimal treatment options for patients without targeted mutations. In the current absence of data on how best to proceed, surgical resectability should be decided in advance at the time of the presentation. Where operable patients in clinical stages II and III are concerned, the current best practice recommends neoadjuvant platinum-based chemotherapy with immunotherapy (neoadjuvant or perioperative) before surgical resection. This approach is preferable to just adjuvant therapy alone. Further, surgical resection should proceed if there is no progression of disease after induction treatment. However, patients with multi-station N2 disease are generally not considered candidates for surgical resection, particularly in the bulky-nodal disease stage because of their poor long-term outcomes. On the other hand, surgical resection can be considered in select cases with non-bulky, 2 to 3 involved N2 stations.

### 6.4. ICI in Patients with Oncogenic Mutation

The benefit of ICI for patients with oncogenic mutations remains largely unclear. Generally, these patients were excluded from clinical trials or not deeply investigated, making this topic quite challenging.

Although EGFR-mutant patients were included in Impower010 and PEARLS, these trials were not designed to specifically evaluate them. The benefit of ICI for patients with oncogenic mutations remains largely unclear. Generally, these patients have been excluded from the clinical trials or simply not considered, which makes it challenging to design a protocol that might improve outcomes [29]. These findings further support the exclusion of this patient subset from neoadjuvant and perioperative ICI-based strategies.

KRAS-driven tumors, particularly those harboring TP53 co-mutations, tend to display a highly inflamed tumor microenvironment, which correlates with improved responses to ICIs [64]. In contrast, the co-occurrence of the STK11 and KEAP1 mutations appear to drive resistance to ICIs, even in tumors with a high tumor mutational burden (TMB), by shaping an immunosuppressive metabolic environment that impairs immune cell activation [65]. These tumors demonstrated augmented sensitivity to immune modulation with the incorporation of a CTLA-4 blockade, delineating a distinct and targetable vulnerability in these subsets [66].

Anyway, a promising possibility towards a more personalized treatment is carried out by next-generation sequencing (NGS) [67].

It would be desirable to have extensive NGS results in all patients to obtain information regarding their prognosis and to select the best treatment for all patients.

### 6.5. Duration of the Treatment

The optimal duration of immunotherapy and dosing intervals must be better elucidated.

The NEOSCORE trial randomized patients with rNSCLC to receive either two or three cycles of the neoadjuvant PD-1 inhibitor sintilimab alongside standard chemotherapy. Although the trial concluded prematurely, it did reveal promising trends, with three cycles yielding improved rates of major and complete tumor responses, albeit falling short of statistical significance.

Findings from advanced NSCLC have shown that the median time to respond to the combination of ICI plus chemotherapy is generally 2 months. Moreover, patients who obtain an objective response following immunotherapy may have proliferation of PD-1 + CD8+ T cells in peripheral blood within 3–4 cycles. As a result, it is plausible that neoadjuvant ICIs for 3–4 cycles could be enough to maximize the immunotherapy effect. Neoadjuvant ICI plus chemotherapy is administered for 3–4 cycles preoperatively in CheckMate-816 [29], KEYNOTE-671 [39], RATIONALE-315 [42], CheckMate 77T [47], AEGEAN [37] and Neotorch [38] investigated four cycles of ICI plus chemotherapy, with three cycles before surgery and one cycle after surgery. All these studies showed significant pCR/MPR rates and EFS benefits.

Also, in the findings from the PACIFIC trial (although in the setting of unresectable stage III NSCLC) and corroborated by real-world evidence, nearly half of the patients initiated on consolidation durvalumab fail to complete the intended year-long immunotherapy regimen. This discontinuation is predominantly attributed to disease advancement or the emergence of adverse events [68]. Furthermore, subsequent studies have demonstrated the effectiveness of shorter durations of consolidation immunotherapy [69,70].

Actually, there are no data to draw final conclusions because of the lack of head-to-head studies evaluating the optimal duration or possible de-escalation strategies such as extending the dosing interval without compromising safety and efficacy.

Future trials are largely awaited to define the safest and most efficacious treatment durations, aiming to mitigate unnecessary costs and adverse events.

### 6.6. Exploring Novel Combination Approaches

The combination of immune checkpoint inhibition with other drugs is under investigation in the perioperative setting. For instance, several published trials have employed dual immunotherapy regimens, such as the NEOSTAR study, which investigated nivolumab plus ipilimumab prior to resection, demonstrating an MPR rate of 50%. Similarly, the NeoCOAST trial exhibited improved MPR with durvalumab combined with one of three novel drugs, namely oleclumab (anti-CD73), monalizumab (anti-NKG2A), and danvartirsent (anti-STAT3 antisense oligonucleotide), compared to durvalumab alone [71].

Ongoing trials like NEOpredict (NCT04205552), assessing neoadjuvant nivolumab in conjunction with the LAG-3 inhibitor relatlimab, and NEOCOAST-2 (NCT05061550), investigating perioperative durvalumab alongside various novel immunochemotherapy combinations including the PD-1/CTLA-4 bispecific volrustomig (NCT05061550), further explore these promising avenues.

We hope to encourage the use of these results in clinical practice.

### 6.7. AI-Driven Precision in Early-Stage NSCLC

Artificial intelligence (AI) is redefining early-stage NSCLC care by merging advanced data layers, such as genomics, transcriptomics, proteomics, and metabolomics, with high-resolution histopathology images to reveal both molecular signatures and tissue architecture changes [72]. These AI-powered platforms not only deliver more objective and reproducible PD-L1 scoring than conventional immunohistochemistry but also automatically quantify TILs, a key indicator of immunotherapy benefit [73]. By feeding refined PD-L1 and TIL measurements or combining PD-L1 with the tumor mutational burden into machine learning models, researchers have achieved superior prediction of ICI responses (AUC 0.77 for TILs/PD-L1 and 0.65 for TMB/PD-L1), notably identifying responders even among PD-L1-negative cases [74]. Thanks to its low cost and seamless integration into digital pathology workflows, AI-driven TIL scoring is poised to become a practical tool for tailoring immunotherapy, pending confirmation in larger, prospective trials.

## 7. Conclusions

In conclusion, the evolving landscape of treatment strategies for rNSCLC underscores the significant role of both neoadjuvant and immunotherapy approaches in enhancing patient outcomes. Neoadjuvant therapies, particularly the integration of platinum-based chemotherapy with ICIs, have demonstrated substantial improvements in MPR rates and OS, positioning them as a promising strategy in the management of early-stage NSCLC. The advent of immunotherapy, both in the neoadjuvant and adjuvant settings, represents a pivotal advancement, offering durable responses and potentially reducing recurrence rates.

The use of ICIs either as monotherapy or in combination with chemotherapy, provided a foundation for ongoing research into personalized treatment approaches. Moreover, the approval of perioperative immunotherapy regimens highlights the importance of a multimodal approach that integrates systemic therapy at various stages of treatment to maximize therapeutic efficacy.

In the current landscape, this drive toward treatment optimization finds strong justification in the perioperative setting, as demonstrated by P.M. Forde’s analysis in 2024 comparing the Checkmate 816 and Checkmate 77T trials, which showed that perioperative nivolumab not only improved EFS but also maintained a generally manageable safety profile.

Future research should continue to explore the optimal sequencing and duration of these therapies, as well as the identification of biomarkers that can guide individualized treatment decisions. By further refining these strategies, the goal is to extend survival, minimize recurrence, and improve the quality of life for patients with rNSCLC.

## Figures and Tables

**Table 1 curroncol-32-00397-t001:** Neoadjuvant immunotherapy studies.

Name	Phase of Study	No. of Participants	Trial Design	Treatment	Stage	Surgical Resection	Primary Endpoints
Forde et al. [20]	II	21	Single arm	Nivolumab 3 mg/kg once every 2 weeks × 2 doses	I-IIIA (AJCC 7th)	95%	MPR: 45%
NCT02904954 Altorki et al. [21]	II	60 (30 + 30)	2 arms	Neoadjuvant: (1) durvalumab once every 3 weeks × 2 doses v (2) durvalumab once every 3 weeks × 2 doses + SBRT optional adjuvant durvalumab once every 4 weeks × 12 months	I-IIIA (AJCC 7th)	86%	MPR Arm 1: 6.7% Arm 2: 53.3%
NEOSTAR study Cascone et al. [22]	II	44	2 arms	Neoadjuvant: nivolumab 3 mg/kg once every 2 weeks v nivolumab 3 mg/kg once every 2 weeks + ipilimumab 1 mg/kg once every 6 weeks × 1 dose	I-IIIA (AJCC 7th)	84%	MPR Arm 1: 22% Arm 2: 38% Undergoing resection (37) Arm 1: MPR 5/21 (24%) Arm 2: MPR 8/16 (50%)
NEOMUN study Eichhorn et al. [23]	II	15	Single arm	Pembrolizumab IV 200 mg once every 3 weeks × 2 doses	II-IIIA (AJCC 7th)	Not reported	MPR: 27%
LCMC3 Chaft et al. [24]	II	181	Single arm	Atezolizumab 1200 mg once every 3 weeks × 2 doses	IB-IIIB (AJCC 8th)	88%	MPR: 20%
IoNESCO study Wislez et al. [25]	II	46	Single arm	Durvalumab 750 mg once every 2 weeks × 3 doses preoperatively	IB-IIIA (AJCC 8th)	93%	MPR: 19%
NEOCOAST Cascone et al. [26]	II	84	Multiarm	Durvalumab 1500 mg once every 4 weeks × 1 dose durvalumab + oleclumab 3000 mg once every 2 weeks × 2 doses durvalumab + monalizumab 750 mg once every 2 weeks × 2 doses durvalumab + danvatirsen 200 mg once daily on days 1, 3, and 5 of week 0, followed once weekly × 4 doses	I-IIIA (AJCC 7th)	92%	MPR: Oleclumab (19%) Monalizumab (30%) Danvatirsen (31%) Durvalumab (11%)

**Table 2 curroncol-32-00397-t002:** Neoadjuvant chemoimmunotherapy studies.

Name	Phase of Study	No. of Participants	Trial Design	Treatment	Stage	Surgical Resection	Primary Endpoints
NCT0271608 Shu C. A. et al. [27]	II	30	Single arm	Neoadjuvant atezolizumab IV 1200 mg on day 1; nab-paclitaxel 100 mg/m^2^ on days 1, 8, and 15; and carboplatin (AUC 5) on day 1 every 21 days for 2–4 cycles	IB-IIIA (AJCC 7th)	96.7%	MPR: 57%
NEOTPD01 Zhao Z. et al. [28]	II	33	Single arm	IV toripalimab 240 mg, carboplatin (AUC 5) + pemetrexed 500 mg/m^2^ or nab-paclitaxel 260 once every 3 weeks for 3 cycles	IIIA or IIIB (AJCC 8th)	91.9%	MPR: 66% pCR: 50%
CheckMate-816 Forde P. M. et al. [29]	III	358	2 arms	Neoadjuvant: Nivolumab 360 mg IV + platinum-doublet chemotherapy once every 3 weeks for 3 cycles v chemotherapy once every 3 weeks for 3 cycles	IB-IIIA (AJCC 7th)	83.2%	EFS: 76.1% at 12 m, 63.8% at 24 m MPR: 24%
NEOSCORE Qiu F. et al. [30]	II	60	2 arms	Neoadjuvant carboplatin AUC 5 and nabpaclitaxel 260 mg/m^2^; pemetrexed 500 mg/m^2^ and sintilimab 200 mg IV once every 21 days for 2–3 cycles, then 1 year of postoperative maintenance with sintilimab	IB-IIIA (AJCC 7th)	91.7%	MPR: 44.1% with 3 cycles 26.9% with 2 cycles
NEOSTAR Platform trial (arm C and D) Cascone T. et al. [31]	II	44	Multi arms	Neoadjuvant nivolumab + CT followed by surgery (arm C) vs. Neoadjuvant ipilimubab + nivolumab + CT followed by surgery (arm D)	IB-IIIA (AJCC 7th)	Arm C 100% Arm D 91%	MPR: 32.1% arm C MPR: 50% arm D

**Table 3 curroncol-32-00397-t003:** Perioperative immunotherapy studies.

Name	Phase of Study	No. of Participants	Treatment	Stage	Surgical Resection	Primary Endpoints (Experimental Group)
IMPOWER 030 Peters S. et al. [42]	III	Not reported	neoadjuvant atezolizumab 1200 mg + double platinum-based CT *4c, followed by surgery and ADJ atezolizumab 1200 mg Q3W *16c	II-III (AJCC 8th)	Not reported	EFS: not reported
NADIM study Provencio M. et al. [43]	II	46	neoadjuvant carboplatin (AUC 6), paclitaxel 200 mg/m^2^, and nivolumab 360 mg Q3W *3c, followed by ADJ nivolumab 240 mg Q2W *4 m and 480 mg Q4W *8 m	IIIA-IIIB (AJCC 7th)	85.4%	PFS 77.1% at 24 m
SAKK 16/14 Rothschild S. I. et al. [44]	II	68	neoadjuvant cisplatin 100 mg/m^2^ and docetaxel 85 mg/m^2^ Q3W *3c, followed by durvalumab 750 mg Q2W *2c and ADJ durvalumab *1y after surgery	IIIA (AJCC 7th)	81%	MPR: 62% pCR: 18%
TOP1501 Tong B. C. et al. [45]	II	35	neoadjuvant pembrolizumab 200 mg Q3W *2c, followed by surgery and ADJ pembrolizumab *4c	IB-IIIA (AJCC 7th)	83.3%	MPR: 28%
LUNGMATE 002 Zhu X. et al. [46]	II	50	neoadjuvant toripalimab 240 mg Q3W *2–4c + carboplatin-based CT, followed by surgery and ADJ toripalimab + carboplatin-based CT	II-III (AJCC 8th)	83%	MPR: 55.6%
AEGEAN Heymach J. V. et al. [37]	III	802	neoadjuvant durvalumab 1500 mg Q3W *4c, followed by surgery and ADJ durvalumab 1500 mg Q4W *12c	II-IIIB (AJCC 8th)	Not reported	pCR: 17.2% EFS: 73.4% at 12 m
NADIM II Provencio M. et al. [41]	II	86	neoadjuvant paclitaxel 200 mg/m^3^ + AUC5 carboplatin + 360 mg of nivolumab Q3W *3c, followed by surgery and nivolumab 480 mg Q4W *6 m	IIIA-IIIB (AJCC 8th)	93%	pCR: 37%
KEYNOTE 671 Wakelee H. et al. [39]	III	797	neoadjuvant pembrolizumab 200 mg Q3W + cisplatin-based CT *4c, followed by surgery and ADJ pembrolizumab 200 mg Q3W *13c	II-III (AJCC 8th)	71.9%	OS: 80.9% at 24 m EFS: 62.4% at 24 m
CHEKMATE 77T Cascone T. et al. [47]	III	458	neoadjuvant nivolumab 360 mg + platinum-based CT Q3W *4c, followed by surgery and ADJ nivolumab 480 mg Q4W *12 m	II-III (AJCC 8th)	Not reported	EFS: 70.2% at 18 m
RATIONALE 315 Yue D. et al. [42]	III	453	neoadjuvant tislelizumab + CT, followed by surgery and ADJ tislelizumab	II-IIIA (AJCC 8th)	80.1%	EFS/OS: not reported
NEOTORCH Shun L. et al. [38]	III	501	neoadjuvant toripalimab 240 mg + platinum-based CT Q3W *3c, followed by surgery and ADJ toripalimab 240 mg Q3W *13c	II-III (AJCC 8th)	Not reported	EFS: 24.4 m MPR: 48.5

## Data Availability

Data are contained within the article.

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
