# Peer review of "Perioperative Chemo/Immunotherapies in Lung Cancer: A Critical Review on the Value of Perioperative Sequences"

_curroncol, 2025, doi:10.3390/curroncol32070397_

Round 1
Reviewer 1 Report
Comments and Suggestions for Authors
Reviewer comments
Unclear sentence (43-44), “According to the eighth edition of…”, is this meant to be connected with a comma instead with the previous sentence (42-43).
Pg 3, Line 118-122. Would argue that it does not appear more advantageous in earlier stages as there are not enough events and it is the opposite of what you have reported for Impower010. This paragraph also appears to be out of place and would recommend merging this in a more cohesive comparison with the next paragraph (Line 123-126).
Pg 3, Line 127-132 – May be helpful to present hypotheses as to there are differential results between adjuvant durvalumab and atezo/pembro (i.e inherent difference in efficacy between PD-L1 and PD-1 inhibitors, trial design, etc.)
Page 3, Line 135-137 – You describe NCCN guidelines including neoadjuvant chemotherapy as a viable option due to randomized trials showing efficacy – however the tables referred to only list neoadjuvant immunotherapy studies and neoadjuvant chemoimmunotherapy studies.
Table 1 and 2 – Would be helpful to add a column indication 1 arm vs 2 arm studies.
Page 5, Line 143-144 – run on sentence “new therapeutic avenues have been explored ICIs have been”
Page 5, Line 178 – change tense of “enhanced” to enhances
Page 6, Line 183-193 – this whole section appears to be repeated and misplaced (should be in the adjuvant section.
Neoadjuvant section – would be helpful to provide a summary and more details of the Checkmate 816 trial which has been practice changing.
Page 8, Line 257-259 – formatting appears different
Page 9, Line 301-304 – would recommend reviewing the matched propensity score data presented at WCLC 2024 which tried to answer the question posed about the benefit of the adjuvant component (Checkmate 77T and Checkmate 816)
Page 10, Line 326-327 – you contradict yourself from page 9 Line 316-317 where you state pCR is considered a valuable surrogate for OS.
Page 10, Line 344-345 – needs a reference
Page 10, Line 354-356 – please review for readability
Page 10, Line 364 – Countries should not be capitalized
Page 10, Line 372-373 – Reference? Is the incidence of TRAES increased in the adjuvant setting or are patients and providers less likely to tolerated toxicities in the curative setting?
Reviewer 2 Report
Comments and Suggestions for Authors
- Innovation: This article systematically reviews the progress of perioperative chemotherapy combined with immunotherapy for resectable non-small cell lung cancer (NSCLC), focusing on the application of immune checkpoint inhibitors (ICIs) in neoadjuvant and adjuvant therapy. By analyzing key clinical trials in the past 20 years (such as CheckMate-816, KEYNOTE-671, AEGEAN, etc.), the article points out:
Breakthrough progress of ICIs: Neoadjuvant or adjuvant chemotherapy combined with ICIs (such as nivolumab, pembrolizumab) significantly improves the pathological complete response rate (pCR) and event-free survival (EFS).
Perioperative strategic advantages: Neoadjuvant therapy can activate the tumor immune microenvironment, and adjuvant therapy can eliminate postoperative residual lesions. The combination of the two further reduces the risk of recurrence.
- Areas that need to be supplemented and modified:
(1) Lack of innovation. Similar reviews have already been published, and it is necessary to further highlight the differences from the existing literature.
(2) In the section of adjuvant therapy, it is recommended to add a similar form.
(3) Biomarker section: ctDNA, PD-L1, TMB, and microbiome were all mentioned, but the analysis was slightly scattered. It is suggested to summarize the predictive value of each biomarker and the current evidence level through a tabular system.
(4) Reference format: Some references have DOI links, while others do not. It is recommended to maintain a uniform format to comply with the journal's norms.
(5) Abbreviations need to be modified: For instance, abbreviations like "NEOADJ" and "ADJ" frequently appear. It is recommended to define them uniformly when they first appear and maintain consistency thereafter to enhance the concisely nature of the language.
Reviewer 3 Report
Comments and Suggestions for Authors. Abstract: good summarization of the article
. Introduction: provides a good contextualization of the theme. Line 42 – 43 – review syntax
. Adjuvant setting:
Consider omitting lines 86-92, since the topic of the article is immunotherapy and if you agree with this omission, reinforce in the introduction that it is out of scope of this article patients EGFR and ALK positive.
. Neoadjuvant setting
Table 1 and 2: consider adding information regarding the AJCC version used in the trials, for stage classification.
Line 143 – 147: review syntax
Line 183 – 193: don’t understand why these two paragraphs are in this section.
. Discussion
Line 264: Three of four regimens? (IMPower 010, PEARLS, Keynote 671 and CheckMate-816).
Line 346: it is missing the reference at the end of the sentence.
. Conclusions
Line 488: consider to omit “ such as nivolumab and pembrolizumab) and refer to ICI in general.
